# Optimizing Apricot Yield and Quality with Biostimulant Interventions: A Comprehensive Analysis

Aurora Cirillo [1,*], Luana Izzo [2], Andrea Ciervo [1], Ivana Ledenko [2], Marco Cepparulo [1], Alfonso Piscitelli [1] and Claudio Di Vaio [1]

1 Department of Agricultural Sciences, University of Naples Federico II, Via Università 100, 80055 Portici, Italy; andreaciervo30@gmail.com (A.C.); marco.cepparulo@unina.it (M.C.); alfonso.piscitelli@unina.it (A.P.); claudio.divaio@unina.it (C.D.V.)
2 Department of Pharmacy, University of Naples Federico II, Via Domenico Montesano 49, 80131 Naples, Italy; luana.izzo@unina.it (L.I.); ivana.ledenko@unina.it (I.L.)
* Correspondence: aurora.cirillo@unina.it

**Abstract:** Biostimulant products are recognized for their ability to improve the agronomic parameters of plants and the qualitative and nutraceutical parameters of fruits and confer greater resistance to plants under abiotic and biotic stress conditions. In our study, we tested three different biostimulants on cultivar "Lady Cot" apricot plants: animal-derived protein hydrolysate, plant-derived protein hydrolysate, and one based on algae to evaluate their effects on improving the agronomic parameters of plants and fruit quality. The product that stood out for providing positive effects was the protein hydrolysate-based product, which increased plant production by 53.80% and yield efficiency by 56.38%. At the same time, it also increased the fruit's diameter growth by approximately 8.3%, showing positive effects on fruit weight as well. The animal-derived protein hydrolysate also reduced acidity by 13.8% and showed a significant increase compared to the control in terms of total polyphenols. Additional research is scheduled to validate these results and ascertain which categories of biostimulant products are most effective in enhancing the agronomic, qualitative, and sensory characteristics of other apricot cultivars.

**Keywords:** protein hydrolysate; fruit quality; *Prunus armeniaca* L.; agronomic parameters; nutraceutical parameters

## 1. Introduction

The apricot tree (*Prunus armeniaca* L.) holds significant prominence among fruit species cultivated globally due to its widespread consumer appeal. Italy stands as the foremost apricot-producing nation worldwide, contributing significantly to both fresh and dried apricot production [1]. Like other perennial fruit trees, the apricot species often encounters various abiotic stresses throughout its lifespan, imposing limitations on crop yield. In addressing this challenge, contemporary fruit tree physiology places a significant emphasis on exploring novel and primarily natural fertilizers known as "Plant biostimulants". These organic substances are designed to enhance nutrient absorption, promote plant development, reduce the reliance on mineral fertilizers, and improve fruit quality [2]. Therefore, biostimulant products, while not being able to replace fertilizers, can be used to reduce their application, enhancing their effectiveness [3]. Recognized for their ability to foster plant growth and fortify plants against harsh conditions, these biostimulants are applied in minimal quantities; in fact, excessive application could disrupt the plant's nutritional equilibrium and lead to adverse effects [3]. They are also acknowledged for their capacity to enhance fruit size, appearance, and other sensory attributes, either through direct influences on fruit growth and development or indirectly by regulating crop load, tree vigor, and canopy architecture [4,5]. Various pomological characteristics play a crucial role in defining the quality attributes of apricot fruits; these traits encompass factors like size, color,

taste, aroma, and firmness [6,7]. Additionally, the quality of apricots is intricately linked to chemical aspects such as sugar and organic acid content, as well as volatile compounds, as observed by Ruiz and Egea [8]. The fruit quality of apricots is a multifaceted concept that involves both physical parameters (including weight, firmness, and skin color) and chemical features (such as soluble solid content, titratable acidity, and pH). This combination is complemented by sensory properties like appearance, texture, taste, and aroma, along with nutritional parameters like phenolic content and antioxidant capacity. These criteria serve as fundamental prerequisites for the fruit industry, and the concept of quality is inherently tied to the overall eating experience of fresh produce [9–11]. From a nutritional standpoint, the apricot epicarp and mesocarp are rich in sugars, organic acids, mineral elements, and vitamins (A, B, and C), along with specific polyphenols [12]. The presence of these nutritional components has consistently captured the attention of numerous scientists due to the potential connections between their dietary intake and the reduced incidence of conditions such as cancer or cardiovascular diseases [13].

The bibliography includes several studies highlighting the use of biostimulant products of various origins to improve the productive and qualitative characteristics of apricots. Tarantino et al. [14] have demonstrated accelerated fruit ripening, enabling a greater quantity of fruit to be harvested at the first harvest and also highlighting an improvement in the nutraceutical component. Other studies [6,15] have also shown that the application of humic acids through soil or foliar spraying positively interferes with the growth, productivity, and fruit quality of the Canino apricot cultivar, observing increased fruit weight, consistency, total soluble solids, and brix-to-acidity ratio, thus promoting better fruit quality.

We hypothesized that the utilization of biostimulants might activate processes to boost the agricultural productivity of crops and improve the qualitative, quantitative, and nutraceutical properties of the fruits in apricot trees. The purpose of our study was to evaluate the effect of biostimulant products of different natures on the "Lady Cot" apricot cultivar to see how such products could improve the agronomic aspects of the plants in terms of fruit set, production, productive efficiency, and how they could enhance the qualitative and nutraceutical characteristics of the fruits.

## 2. Materials and Methods

### 2.1. Site and Biostimulants

This study was conducted at *Azienda Gallo* (39°43′38.5″ N 16°24′18.4″ E), a cooperative agricultural company in Castrovillari, Calabria (South Italy), with the primary aim of investigating the effects of three biostimulant products, all belonging to the *Diachem* S.p.A. company (BG), on the cultivation of apricots (*Prunus armeniaca* L.) of eight-year-old trees of the Lady Cot cultivar grafted on Montclar rootstock. The plants were trained to vase systems and spaced 5 m between the rows and 4 m within the rows with a drip irrigation system (16 L per plant).

The tested products included the following:

(1) A biostimulant based on an extract derived from the seaweed Ascophyllum nodosum collected in the Atlantic Ocean (Nova Scotia, Canada), commercially known as Enerleaf. It was applied by foliar application at a dose of 400 mL/100 L of water.

The composition also includes organic carbon (C) at 2.5% (25 g/L), alginic acid at 10% (10 g/L), and mannitol at 0.4% (4 g/L).

(2) A biostimulant based on an organic nitrogen fertilizer obtained through controlled enzymatic hydrolysis of selected plants with a high content of free amino acids, oligopeptides, enzymes, vitamins, elicitors, and substances with hormone-like functions, commercially known as Aminomix Vegetal. It was applied by foliar application at a dose of 400 mL/100 L of water.

The composition also includes total nitrogen (N) at 6%, organic nitrogen (N) at 6%, water-soluble potassium oxide ($K_2O$) at 6%, organic carbon (C) at 18%, and amino acids at 38%.

(3) A biostimulant based on a nitrogen-rich organic fertilizer and fluid suspension of meat by-products, commercially known as Aminozime Ultra. It was applied by foliar application at a dose of 200 mL/100 L of water.

The composition also includes total nitrogen (N) at 3% (38.25 g/L), including soluble organic nitrogen at 3% (38.2 g/L), organic carbon (C) at 10% (127.5 g/L, equivalent to total amino acids at 18.75% (239.1 g/L)), chelated micronutrients at 0.5% (6.3 g/L), vitamins (B1, B2, B5, PP, C, and H1) at 0.1% (1.2 g/L), polysaccharides at 2% (25.5 g/L), and plant-origin auxins.

(4) The three selected commercial products were compared with a control group, where only water was administered.

The study was organized as a complete experimental design.

It involved the arrangement of plants in a completely randomized manner (Figure 1) with four treatments implemented at key stages of the vegetative season, precisely before flowering (16 March 2023), fruit set (1 April 2023), post-fruit set (26 April 2023), and fruit development (23 May 2023), corresponding, respectively, to the following stages of the BBCH scale: 57, 71, 75, and 79 [16]. The minimum, maximum, and average temperature data recorded during the growing season were collected from the meteorological station of Castrovillari (CS), located in the Regional Environmental Protection Agency (Calabria). The irrigation water used during the plant growth period was made available by the reclamation consortium of the Piana di Sibari and the Media Valle di Crati (CS) on a 3-day rotation.

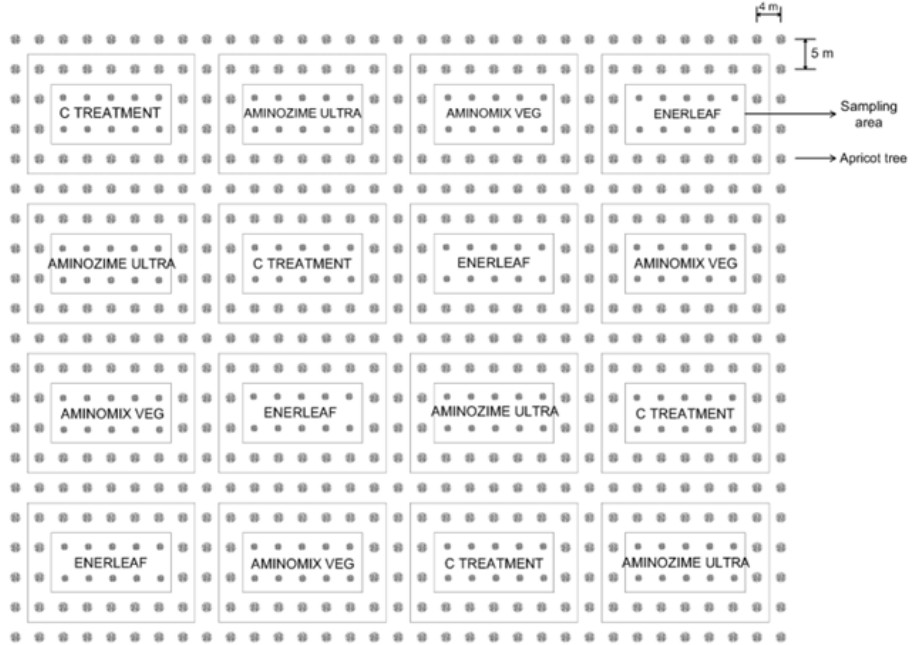

**Figure 1.** Design completes of the experimental layout.

### 2.2. Biometric Analysis and Physico-Chemical Analysis of Fruits

Field measurements included the count of flowers per fruiting branch and the count of fruits attached per fruiting branch to determine the percentage of fruit set. Additionally, observations were made on the growth of fruit diameter on the plant and the overall plant production. Subsequent calculations involved assessing productive efficiency, determined by the ratio of plant production to the cross-sectional area of the trunk (TCSA). TCSA was calculated using the standard formula (girth$^2$/4$\pi$), and the study also investigated the production obtained at the first harvest.

In the laboratory, a sample of 100 fruits per treatment underwent the following analyses: commercial caliber, polar, equatorial, and transversal diameter of the fruits (mm), determined using a digital vernier caliper (Mitutoyo, Kawasaki, Japan), fruit weight (g), determined with an electronic digital balance (Precisa Instruments AG, model XB220A, Dietikon, Switzerland), pulp hardness (kg/0.5 cm$^2$), determined with an EFFEGI manual penetrometer with an 0.5 cm$^2$ tip on two sides opposite the fruit, epicarp color, measured with a colorimeter using color coordinates *L, *a, and *b, subsequently used to calculate Chroma (*a$^2$ + *b$^2$)$^{(1/2)}$. The level of total soluble solids (TSSs), represented in °Brix, was assessed using a digital refractometer (Atago, model PR-101a, Tokyo, Japan), while the pH was gauged utilizing a digital pH meter (Crison Instruments, model GLP 21, Barcelona, Spain). Total acidity (TA) was ascertained through an acid–base titration, where the solution underwent titration with a 0.1 N standard solution of sodium hydroxide, and the result was expressed as grams of citric acid per 100 milliliters. Subsequently, the TSS/TA ratio was computed. The analysis of fruit nutraceutical parameters encompassed antioxidant activity through ABTS, DPPH, and FRAP assays, and individual polyphenols and total polyphenols were determined using UHPLC-Orbitrap instrumentation.

### 2.3. Polyphenol Extraction

A freeze-dried sample (1 g) was extracted with 5 mL of MeOH:H$_2$O (80:20, *v/v*) containing 0.1% FA, in accordance with the procedure reported by Cirillo et al. [11]. The mixture was mixed (ZX3, VEPL Scientific, Usmate, Italy) for 3 min, subsequently sonicated (LBS 1, Zetalab srl, Padua, Italy) for 10 min, and agitated for a further 10 min on a digital orbital shaker (SKO-D XL, ARGOlab, Arezzo, Italy). The mixture was then centrifuged for 5 min at 4 °C, 5000 rpm, the supernatant was collected, and the procedure was performed another time using 5 mL of the MeOH:H$_2$O (80:20, *v/v*) mixture containing 0.1% FA. The overall supernatant was combined and filtered (0.22 μm filter) for the UHPLC Q-Orbitrap HRMS analysis.

### 2.4. Chemical Characterization of Polyphenols through UHPLC Q-Exactive Analysis

An Ultra-High-Pressure Liquid Chromatograph coupled with Q-Exactive (Thermo Fisher Scientific, Waltham, MA, USA) was employed in the qualitative–quantitative analysis of polyphenols. Chromatographic separation was performed using a Kinetex 1.7 μm F5 column (50 × 2.1 mm, Phenomenex, Torrance, CA, USA). The mobile phase was water (A) and methanol (B) with 0.1% FA, and the flow rate and the injection time were set at 0.5 mL/min and 5 μL, respectively. The gradient elution begins at 0% B and increases to 40% B in one minute, 80% B in one minute, and 100% B in three minutes. To allow for column re-equilibration, the gradient was held at 100% B for 4 min, then dropped to 0% B for 2 min, and, finally, maintained at 0% for an additional 2 min. The instrument was operated in both the negative and positive ion modes, and two scan events were set. In the full MS mode, a scan range of 80–1200 *m/z*, a resolution power of 70,000 full width at half maximum (FWHM), an automatic gain control target of 1 × 10$^6$, an injection time set to 200 ms, and a scan rate of 2 scan/s were set. The following characteristics were used: a sheath gas pressure of 18, an auxiliary gas of 3, a spray voltage of 3.5 kV, a capillary temperature of 320 °C, a S-lens RF level of 60, and a heater temperature of 350 °C for the auxiliary gas. In the AIF scan event, the following characteristics were set: scan range of 80–120 *m/z*, mass resolving power of 17,500 FWHM, ACG target of 1 × 10$^5$, isolation window of 5.0 *m/z*, retention time of 30 s, maximum injection time of 200 ms, and scan time of 0.10 s. Product ion spectra were obtained by varying the collision energy within the range of 10 to 60 eV. Both the fragments and the molecular ion were identified and confirmed with a mass tolerance of 5 ppm [17]. The Xcalibur software v. 3.1.66.10 was used for data processing and analysis (Xcalibur, Thermo Fisher Scientific, Waltham, MA, USA).

*2.5. Total Phenolic Content Analysis*

The Folin–Ciocalteu technique was used to calculate the total phenolic content, in accordance with the procedure reported by Izzo et al [17]. In short, 125 µL of the extract, 500 µL of deionized water, and 125 µL of Folin–Ciocalteu reagent 2 N were combined. The tube was mixed and then left in the dark for six minutes. Subsequently, 1.25 mL of a 7.5% sodium carbonate solution and 1 mL of deionized water were added. For ninety minutes, the reaction mixture was maintained in the dark. Ultimately, the absorbance at 760 nm was recorded using a spectrophotometer. Gallic acid equivalents (GAEs) per gram of dry weight sample were used to express the results.

*2.6. Antioxidant Activity*

Three distinct colorimetric tests were used to assess the antioxidant activity: the Free Radical-Scavenging Assay (DPPH), the Radical Cation Scavenging Assay (ABTS), and the Ferric Reducing Antioxidant Power (FRAP) [18]. For evaluating the ferric reducing antioxidant power, ferric chloride solution (20 mM), acetate buffer (0.3 M; pH 3.6), and TPTZ solution (10 mM) were mixed in a 1:1:10 (*v/v/v*) ratio to form the FRAP reagent. The assay was carried out on 150 µL of an opportunity-diluted sample quickly mixed with 2.850 mL of FRAP reagent. After four minutes, the absorbance value at 593 nm was recorded. For evaluating the free radical-scavenging activity, deionized water was used to dissolve ABTS diammonium salt to a concentration of 7 mM. Then, 44 µL of potassium persulfate solution (2.45 mM) was added. For sixteen hours, the solution was kept at room temperature and in the dark. Subsequently, the ABTS solution was diluted with ethanol until its absorbance value at 734 nm was 0.70 (±0.02). Next, 1 mL of ABTS solution was mixed with 0.1 mL of the suitably diluted sample. After waiting 2.5 min, the absorbance was measured at 734 nm. For evaluating the total free radical-scavenging activity, 4.0 mg of DPPH was dissolved in 10 mL of MeOH, and the solution was then diluted to reach an absorbance of 0.90 (±0.02) at 517 nm. Then, 1 mL of the working solution and 200 µL of sample extract were combined to perform the experiment. Antioxidant activities were expressed as mmol Trolox Equivalents/kg of dry weight sample.

*2.7. Statistical Analysis*

After verifying both that all the measures were normally distributed through the Kolmogorov–Smirnov test and the homogeneity of the variances through the Levene's test, a one-way ANOVA was applied for each measure in order to verify statistical significance differences among treatments and the control group. Duncan's multiple range test ($p = 0.05$) was performed because it is a statistical method used in agricultural research to evaluate the effects of treatments. It is particularly useful for identifying optimal treatments, optimizing agricultural practices, and supporting crop genetic research [19]. Data were analyzed with Microsoft Excel and IBM® SPSS Statistics, Package 6, version 23.0. A heatmap was generated using the Clustvis online tool, and matrix values were normalized as ln (x + 1) with Euclidean distance and complete linkage.

**3. Results and Discussion**

*3.1. Effect of Biostimulants on Biometric and Physico-Chemical Traits of Fruits*

In the present study conducted in Castrovillari (CS) in Calabria (South Italy), we investigated the effect of three biostimulant products based on hydrolyzed proteins of animal origin, hydrolyzed proteins of vegetable origin, and algae-based products on the improvement of certain agronomic parameters of plants, fruit quality, and the nutraceutical parameters of the fruits. In Figure 2, the average minimum and maximum temperature data recorded during the growing season are reported. From the graph, it can be observed that significant temperature fluctuations occurred throughout the season, starting from minimum temperatures of 2.2 °C in March to 14.4 °C in May, while maximum temperatures peaked at 26 °C in May and reached minimum values of 10 °C in April.

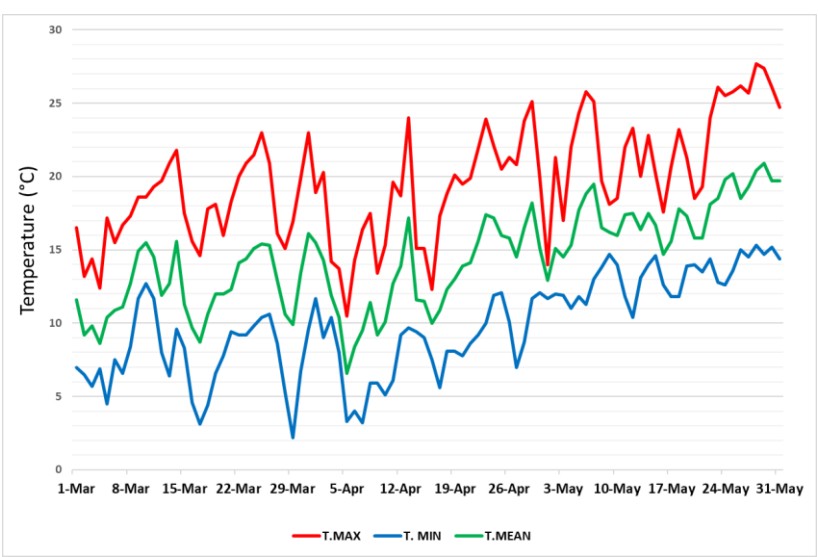

**Figure 2.** Trends of maximum, minimum, and average temperatures recorded throughout the growing season.

Table 1 shows the production parameters of treated plants and control.

**Table 1.** Effect of the three biostimulant products and control on production per plant (kg), yield efficiency (kg/cm$^2$), and on the first harvest (kg/pt).

| Treatments | Production Plant (kg) | Yield Efficiency (kg/cm$^2$) | First Harvest (kg/pt) |
|---|---|---|---|
| Control | 31.65 ± 3.17 [b] | 0.94 ± 0.10 [b] | 8.59 ± 1.62 |
| Enerleaf | 34.41 ± 3.17 [b] | 1.07 ± 0.10 [b] | 11.20 ± 1.62 |
| Aminomix Vegetal | 36.67 ± 3.17 [b] | 1.21 ± 0.10 [b] | 14.62 ± 1.62 |
| Aminozime Ultra | 48.68 ± 3.17 [a] | 1.47 ± 0.10 [a] | 10.15 ± 1.62 |
| Significance | * | * | ns |

Values are mean ± standard error. Asterisks indicate a significant effect of biostimulant treatments according to ANOVA (ns = not significant; * = $p < 0.05$). Different letters indicate significant differences based on Duncan's test ($p = 0.05$).

Fruit set results did not show statistically significant differences between treatments, while statistically significant differences were found for production per plant and yield efficiency. Aminozime Ultra application showed an increase in production per plant of about 53.80% compared to the control, and at the same time, this treatment led to an increase in yield efficiency of 56.38% compared to the control. Regarding the first harvest values, although not statistically significant, the plants treated with the product Aminomix Vegetal show a slight advancement in fruit harvest, thus highlighting a potential advance in the ripening of the fruits. It is known that the enhancement in terms of production and yield is due to an increase in the absorption of macro- and micronutrients [20]. One potential explanation for the increase in yield observed with the foliar application of this biostimulant could be attributed to the presence of a complex of amino acids, complexed microelements, vitamins, and auxins of plant origin found in the product. It is widely recognized that these important molecules can be readily absorbed by the leaf surface, where they act as signaling agents, ultimately enhancing plant growth and, consequently, boosting crop productivity [21]. In the literature, there are several studies confirming the positive effect of biostimulant products on the growth, yield, and overall fruit quality of peach and apple trees [1], as well as on the "Canino" apricot variety [1,15]. Additionally, these studies have shown that the foliar application of seaweed extract enhances root development in grapevines and strawberries [22].

The analysis of fruit diameter growth throughout the vegetative season revealed a significant increase in fruit development following the application of two protein hydrolysates, namely Aminozime Ultra (+8.3%) and Aminomix Vegetal (+3.0%), compared to the control (Figure 3A). The fruit growth rate (Figure 3B) shows a highly elevated development increase from April to mid-May, reaching approximately 16 mm, after which it tends to decrease dramatically. The dimension of fruit is a crucial determinant linked to the perception of superior fruit quality, bearing significant importance not only for consumers but also in terms of commercial value [23]. Nevertheless, in the case of most fruits cultivated commercially, their size often falls short of meeting market expectations. As a result, farmers tend to prioritize enlarging fruit size, even if it means sacrificing the overall quantity of fruits produced with thinning. Nonetheless, the precise mechanism underlying the increase in fruit yield and size following biostimulant treatment remains unclear and poorly understood. To date, several researchers have linked these effects to potential enhancements in plant enzymatic systems due to the chelating metal activity of biostimulants or to their auxin- and gibberellin-like properties [24,25], as may have happened in our case study. The positive results obtained with the animal protein hydrolysate, both on fruit diameter growth and production, are consistent with a previous study we conducted on apple trees, confirming that this biostimulant product on *Malus Domestica* increased production by about 15% compared to the control. In both years of the test, it significantly increased fruit diameter [4].

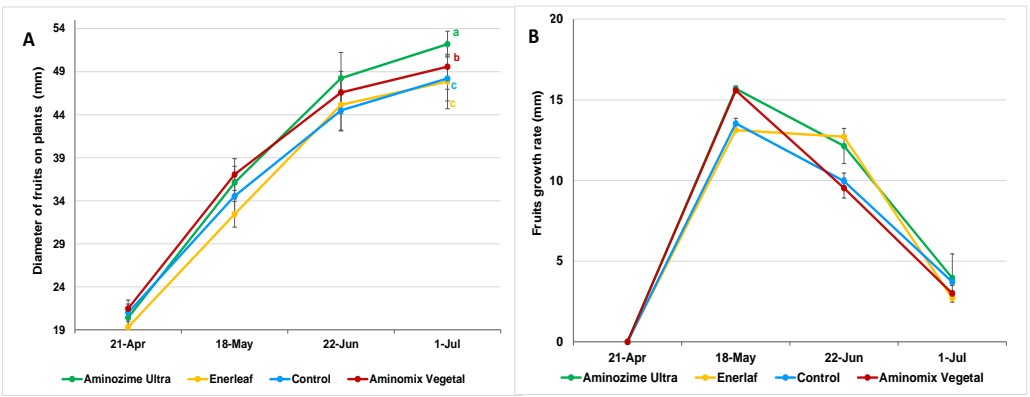

**Figure 3.** Effects of the three biostimulants and control on the growth of the fruits detected on the plants (**A**) and of the growth speed of the fruits (**B**). Values are mean ± standard deviation. Different letters indicate significant differences based on Duncan's test (*p* = 0.05).

Table 2 shows the physico-chemical parameters of the fruits treated with the three biostimulants compared to the control.

The fruit weight was found to be higher with Aminozime Ultra, showing an increase of 28% compared to the control. Also, the polar, equatorial, and transverse diameters confirmed greater fruit development with Aminozime Ultra. However, no significant differences were observed in terms of pulp hardness and fruit sugar content (measured as °Brix) with the treatments applied. Regarding the titratable acidity of the fruits, significant reductions were highlighted with Aminozime Ultra (13.8%) and Enerleaf (14.4%) compared to the control, consequently improving the fruit's palatability for consumers. Our results are consistent with those reported by Mosa et al. [26], where the use of three biostimulant products showed a reduction in acidity compared to the control in apples; in our case, a higher value of the TSS/TA ratio was observed, indicating a potential greater appreciation by the consumer. The firmness of the pulp did not show significant differences among the various treatments, but a 10% reduction was highlighted with the application of Aminomix Vegetal, confirming a higher percentage of the first harvest compared to the other treatments, as previously explained. In the scientific literature, there are several studies confirming that the use of biostimulant products has positive effects on the precocity of fruit ripening, which translates into positive effects on fruit commercialization during periods of higher

demand from the market [27–29]. Regarding the distribution of fruits in commercial sizes (Figure 4), Aminozime Ultra showed significantly higher percentage values for fruits in the upper B and C classes. In particular, letters B, C, D, and E correspond to different fruit diameters, specifically B diameter ranging from 61–67 mm, C diameter ranging from 56–61 mm, D diameter ranging from 51–56 mm, and E diameter being less than 56 mm.

**Table 2.** Effects of the three biostimulants and control on the physical-chemical parameters of the fruits.

| Treatments | Fruit Weight (g) | Polar Diameter (mm) | Equatorial Diameter (mm) | Transversal Diameter (mm) | Firmness (kg × 0.5 cm²) |
|---|---|---|---|---|---|
| Control | 61.00 ± 1.20 [c] | 48.92 ± 0.35 [c] | 46.03 ± 0.40 [c] | 50.50 ± 0.40 [c] | 2.33 ± 0.12 |
| Enerleaf | 59.20 ± 1.20 [c] | 49.17 ± 0.35 [c] | 45.88 ± 0.40 [c] | 50.33 ± 0.40 [c] | 2.37 ± 0.12 |
| Aminomix Vegetal | 65.62 ± 1.20 [b] | 50.55 ± 0.35 [b] | 48.07 ± 0.40 [b] | 51.93 ± 0.40 [b] | 2.09 ± 0.12 |
| Aminozime Ultra | 78.62 ± 1.20 [a] | 53.52 ± 0.35 [a] | 51.51 ± 0.40 [a] | 55.54 ± 0.40 [a] | 2.11 ± 0.12 |
| Significance | *** | *** | *** | *** | ns |

| Treatments | TSS (°brix) | TA (g/L citric acid) | TSS/TA | pH |
|---|---|---|---|---|
| Control | 12.67 ± 0.31 | 18.44 ± 0.72 [a] | 0.72 ± 0.34 | 3.89 ± 0.09 |
| Enerleaf | 12.44 ± 0.31 | 15.78 ± 0.72 [b] | 0.78 ± 0.34 | 3.89 ± 0.09 |
| Aminomix Vegetal | 12.89 ± 0.31 | 17.33 ± 0.72 [ab] | 0.75 ± 0.34 | 4.00 ± 0.09 |
| Aminozime Ultra | 13.22 ± 0.31 | 15.89 ± 0.72 [b] | 0.82 ± 0.34 | 3.89 ± 0.09 |
| Significance | ns | * | ns | ns |

Values are mean ± standard error. Asterisks indicate a significant effect of the biostimulant treatments according to ANOVA (ns = not significant; * = $p < 0.05$; *** $< 0.001$). Different letters indicate significant differences based on Duncan's test ($p = 0.05$).

The increase in fruit size is likely due to the composition of the biostimulant Aminozime Ultra, which, among other elements, contains auxins, hormones involved in growth stimulation through cell elongation [30]. The significant increase in fruit size is essential as it enhances their appeal to consumers, resulting in higher commercial value compared to smaller fruits [6].

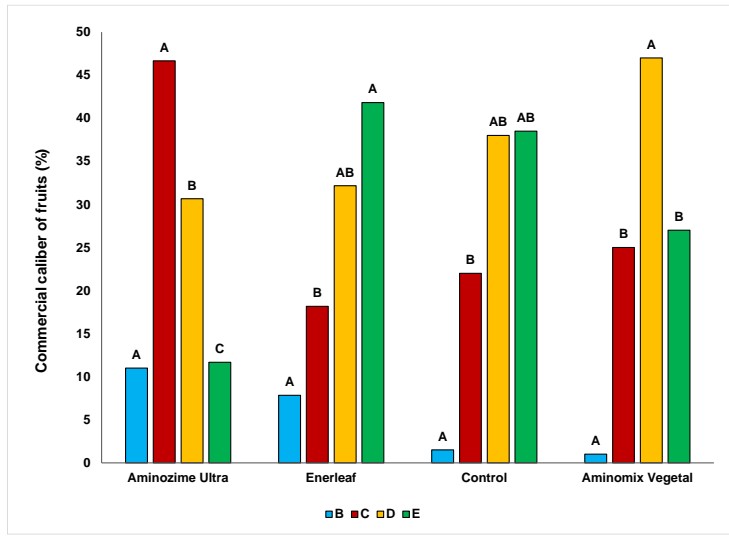

**Figure 4.** Effect of the three biostimulants and control on commercial fruit size. Different letters indicate significant differences based on Duncan's test ($p = 0.05$).

### 3.2. Effects of Biostimulants on Color Fruits and Nutraceutical Parameters

The coloration of apricot fruit serves as a valuable indicator of both ripeness and carotenoid levels [31,32] while also acting as a decisive factor in shaping consumer preferences. In retail settings, shoppers tend to gravitate toward varieties boasting visually pleasing hues when making their buying choices. Conversely, within industrial contexts, color assumes a critical role in the sorting, grading, and routing of fruit through subsequent processing stages [33]. Among the various positive effects exerted by biostimulant products on fruit quality is the enhancement of fruit coloration. Indeed, studies reported in the literature have demonstrated the significant role of biostimulants in improving the color efficiency of fruit peels [7,34].

Regarding the evaluation of the skin color of the "Lady Cot" apricot cultivar in our study, the treatments employed did not increase the intensity of the red color; on the contrary, the use of Aminomix Vegetal led to a slight reduction in red overcoloring. The analyzed cultivar is highly valued in the market due to its appearance and coloration; indeed, it exhibits a bright orange background color with red overcoloring extending over 30–40% of the surface. The values of the *L (luminosity), *a (red intensity), *b (yellow intensity), and Chroma parameters are reported in Table 3.

**Table 3.** Effects of the three biostimulants and control on the coloration of fruit epicarp, detected using a colorimeter.

| Treatments | *L | *a | *b | Chroma |
|---|---|---|---|---|
| Control | 46.23 ± 0.62 [b] | 54.71 ± 1.87 [a] | 29.87 ± 4.88 | 65.34 ± 47.89 |
| Enerleaf | 47.63 ± 0.62 [ab] | 51.03 ± 1.87 [ab] | 27.65 ± 4.88 | 157.87 ± 47.89 |
| Aminomix Vegetal | 48.85 ± 0.62 [a] | 46.78 ± 1.87 [b] | 33.25 ± 4.88 | 59.13 ± 47.89 |
| Aminozime Ultra | 48.87 ± 0.62 [a] | 51.18 ± 1.87 [ab] | 33.51 ± 4.88 | 63.11 ± 47.89 |
| Significance | ** | * | ns | ns |

Values are mean ± standard error. Asterisks indicate a significant effect of biostimulant treatments according to ANOVA (ns = not significant; * = $p < 0.05$; ** = $p < 0.01$). Different letters indicate significant differences based on Duncan's test ($p = 0.05$).

In general, the biostimulants used did not alter the red coloration of the epicarp in our case study, since the Lady Cot cultivar is already distinguished by an intense red overcolor.

### 3.3. Antioxidant Activity and Identification and Quantification of Apricot Bioactive Compounds

Among the objectives of the present study was to furnish insights into enhancing composition in polyphenolic content and conduct a thorough assessment of antioxidant activity in extracts by utilizing biostimulant products. Thanks to their composition, apricots and their by-products are very important for human health [11]. They offer various beneficial effects, including protection against cancer, cardiovascular disease, atherosclerosis, and age-related illnesses, as well as safeguarding the kidneys and liver [35]. Apricots contain numerous secondary metabolites, many of which act as antioxidants. Among these, polyphenols and carotenoids are the most prevalent phytochemicals, influencing both the color and flavor of the fruit. Numerous studies have analyzed the quality, composition, and biochemistry of apricots [35–39], and in the bibliography, there are also studies that highlight the improvement of the nutracutic aspects of fruits with the use of biostimulant products [7,11]. Indeed, beyond addressing agronomic requirements aimed at optimizing yield and enhancing crop quality, there's a growing consumer demand for products rich in nutritional and nutraceutical value [40].

As reported in Tables 4 and 5, the content of polyphenols and the antioxidant activity of fruits treated with biostimulants increase significantly.

**Table 4.** Effect of the biostimulant products and control on the chemical characterization of the main polyphenols in the investigated extracts.

| Polyphneols (µg/g) | Control | Aminozime Ultra | Aminomix Vegetal | Enerleaf | Significance |
|---|---|---|---|---|---|
| Quinic acid | 346.42 ± 20.77 [b] | 415.74 ± 20.77 [a] | 381.08 ± 20.77 [b] | 389.75 ± 20.77 [a] | *** |
| Protocatechuic acid | 1.51 ± 0.15 [b] | 2.53 ± 0.15 [a] | 2.01 ± 0.15 [b] | 2.66 ± 0.15 [a] | * |
| Caffeic acid | 16.60 ± 0.86 | 18.34 ± 0.86 | 16.72 ± 0.86 | 19.15 ± 0.86 | ns |
| Epicatechin | 30.34 ± 1.54 | 26.06 ± 1.54 | 24.68 ± 1.54 | 27.24 ± 1.54 | ns |
| Chlorogenic acid | 641.14 ± 29.97 | 663.93 ± 29.97 | 539.74 ± 29.97 | 649.41 ± 29.97 | ns |
| Catechin | 76.84 ± 4.23 | 82.42 ± 4.23 | 85.65 ± 4.23 | 102.55 ± 4.23 | ns |
| p-coumaric acid | 10.10 ± 0.67 [b] | 12.39 ± 0.67 [ab] | 13.92 ± 0.67 [ab] | 16.45 ± 0.67 [a] | * |
| acid Syringic acid | 24.38 ± 3.18 [c] | 65.53 ± 3.18 [b] | 68.63 ± 3.18 [b] | 104.92 ± 3.18 [a] | *** |
| Ferulic acid | 58.64 ± 2.52 [b] | 56.61 ± 2.52 [b] | 62.79 ± 2.52 [b] | 79.93 ± 2.52 [a] | * |
| Naringin | 0.99 ± 0.05 | 0.90 ± 0.05 | 0.94 ± 0.05 | 1.23 ± 0.05 | ns |
| Rutin hydrate | 508.06 ± 28.69 | 544.14 ± 28.69 | 383.58 ± 28.69 | 557.87 ± 28.69 | ns |
| Quercetin 3β-glucoside | 4.82 ± 0.33 | 4.58 ± 0.33 | 3.21 ± 0.33 | 4.52 ± 0.33 | ns |
| Kaempferolo 3-O-glucoside | 0.56 ± 0.04 | 0.67 ± 0.04 | 0.41 ± 0.04 | 0.64 ± 0.04 | ns |
| isorhamnetin-3-rutinoside | 1.89 ± 0.08 | 1.57 ± 0.08 | 1.59 ± 0.08 | 2.00 ± 0.08 | ns |
| Luteolin 7-glucoside | 0.85 ± 0.06 | 0.91 ± 0.06 | 0.56 ± 0.06 | 0.84 ± 0.06 | ns |
| Myricitrin | 4.64 ± 0.31 | 4.09 ± 0.31 | 3.19 ± 0.31 | 4.52 ± 0.31 | ns |
| Sum of Total (poly) phenols | 1727.75 ± 60.30 [b] | 1900.40 ± 60.30 [a] | 1588.7 ± 60.30 [c] | 1963.67 ± 60.30 [a] | * |

Values are mean ± standard error from three independent determinations (µg/g). Asterisks indicate a significant effect of biostimulant treatments according to ANOVA (ns = not significant; * = $p < 0.05$; *** = $p < 0.001$). Different letters indicate significant differences based on Duncan's test ($p = 0.05$) on the rows.

**Table 5.** Antioxidant activity evaluated by FRAP, DPPH, and ABTS assays and total phenolic content evaluated by FOLIN.

| Treatments | DPPH | ABTS | FRAP | FOLIN |
|---|---|---|---|---|
| | | mmol trolox/kg | | mg/g |
| Control | 9.47 ± 1.06 | 19.15 ± 1.63 | 15.73 ± 2.25 | 5.32 ± 0.64 |
| Aminozime Ultra | 9.95 ± 1.06 | 18.72 ± 1.63 | 16.57 ± 2.25 | 5.64 ± 0.64 |
| Enerleaf | 9.55 ± 1.06 | 17.72 ± 1.63 | 15.33 ± 2.25 | 5.48 ± 0.64 |
| Aminomix Vegetal | 8.15 ± 1.06 | 14.73 ± 1.63 | 11.50 ± 2.25 | 4.84 ± 0.64 |
| Significance | ns | ns | ns | ns |

Values are mean ± standard error from three independent determinations (ns = not significant).

Aminozime Ultra has an increase of 22.7, 67.5, and 168.8% for *p*-coumaric acid, protocatechuic acid, and syringic acid, respectively (Table 4). Similar results were shown for Enerleaf. In this case, ferulic acid, *p*-coumaric acid, Protocatechuic acid, and syringic acid increased by 36.3, 62.9, and 330.4% compared to the control, respectively. Aminomix Vegetal reported an increase of 181.5% for syringic acid (Table 4).

Antioxidant activity (Table 5) was higher with samples treated with Aminozime Ultra, showing an increase of 5.1% and 5.3%, respectively, for the DPPH and FRAP tests compared to the control. In line with these results, the TPC also registered an increase of 6% for Aminozime Ultra compared to the control. Evaluating the content of polyphenols, the treatment with Aminozime Ultra (+10%) and Enerleaf (+13.7%) improved the total polyphenol content.

Plant biostimulants have become a novel production tool over the past ten years. They have the potential to improve crop tolerance to various abiotic stressors, such as drought, salinity, extreme temperature, and radiation, and to improve the final quality of food products [41]. Parrado et al. [42] investigated the effects of soil applications of protein hydrolysate of vegetal origin (corn, sorghum, and carob) on must composition and color for the wine grape cultivar "Tempranillo". Remarkably, the outcomes showed that, when compared to the untreated controls, the biostimulant application caused increases in required total polyphenols and anthocyanins, respectively, by 28% and 227%. Moreover, Cirillo et al. [43] studied the impact of three biostimulants on oil yield, the number of drupes

produced per plant, and the nutritional value of olive oil and drupes for "Racioppella" cultivar trees. The use of biostimulants helped to improve fundamental metrics. All three biostimulants had a beneficial effect by raising the polyphenol content of olive oil from 7.8 to 32.2% compared to the control. When applied to plants, biostimulants can enhance their growth and development, as well as the quality and nutritional worth of their output. They can also increase nutrient uptake, yield, and water content [44]. However, there is still more to be achieved in this area; therefore, a study is still needed.

### 3.4. Heatmap and Cluster Analysis

To obtain an overview of the achieved results, a cluster analysis was conducted for all the aforementioned measured parameters, and the obtained clusters were visualized with the help of a heat map. The different treatments lead to the formation of three clusters, where the vegetable protein hydrolysate product Aminomix Vegetal, the control group, and the last cluster, which includes the animal-origin protein hydrolysate Aminozime Ultra and the algae extract-based product Enerleaf, are divided. Regarding the Aminomix Vegetal treatment, from the heatmap, a high correlation with the percentage of the first harvest, expressed by intense red coloring, is clearly visible. As explained earlier, this could be indicative of a faster induction of the maturation process. The control group, on the other hand, showed a higher correlation with some polyphenols identified by UHPLC-Orbitrap instrumentation, such as Epicatechins. The last cluster, comprising the other two biostimulants, shows a very high correlation with the application of the Aminomix Ultra product, fruit weight, and size, highlighting positive effects on some agronomic parameters such as plant yield and productivity efficiency. The Enerleaf product, instead, exhibited a high correlation expressed in terms of pulp hardness, catechin, ferulic acid, and naringin (Figure 5).

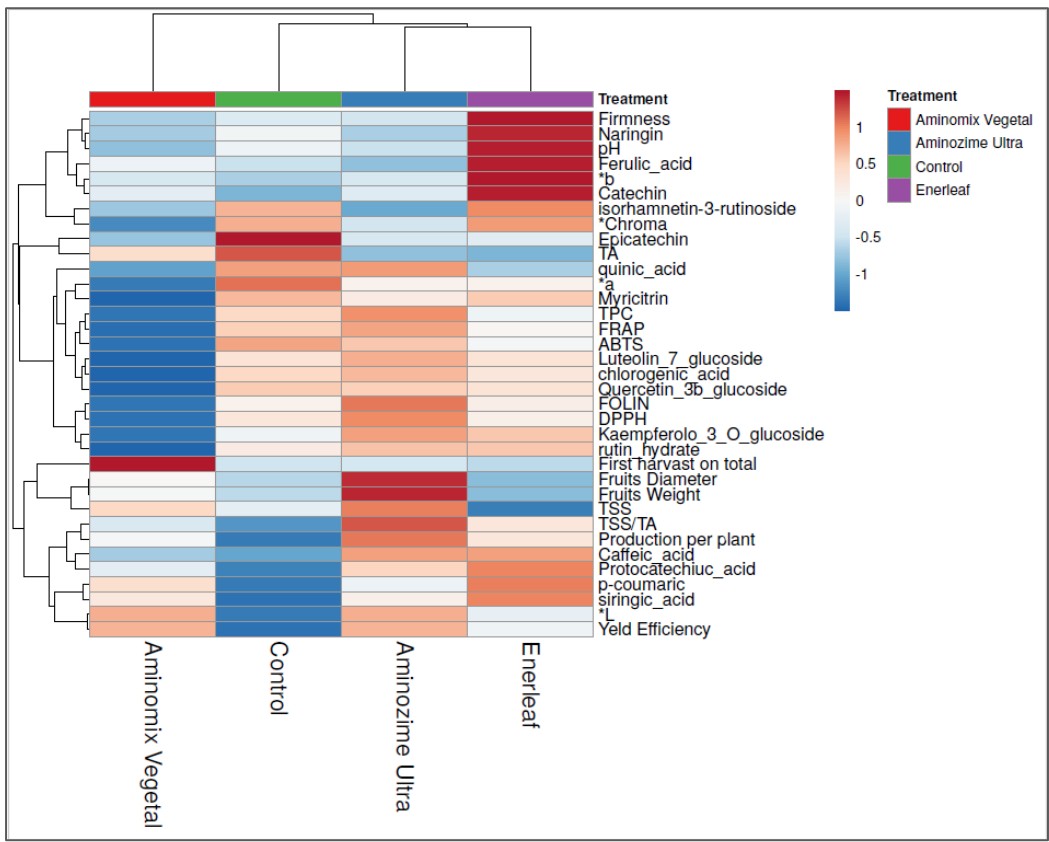

**Figure 5.** Heat map of the cluster analysis that provides a visual summary of both agronomic parameters related to apricot plants and the qualitative and quantitative nutraceutical characteristics of the fruits.

### 4. Conclusions

Our research has shown that the effects of various types of biostimulant products can enhance both the agronomic parameters of apricot plants and the qualitative, quantitative, and nutraceutical characteristics of the fruits. Among the biostimulants tested, Aminozime Ultra, formulated with animal-origin protein hydrolysates, demonstrated positive effects on the analyzed parameters of the Lady Cot cultivar. Significant improvements were recorded, including a 36.2% increase in plant yield, a 35.4% enhancement in productivity efficiency, and an 8.3% increase in fruit diameter, leading to higher commercial grades, particularly in classes B and C. Additionally, a 13.8% reduction in fruit acidity was observed, indicating a positive impact on organoleptic quality. Regarding the nutraceutical characteristics of the fruits, Aminozime Ultra resulted in a 10% increase in the total polyphenol content.

Further research should be carried out to determine which groups of biostimulants are most suitable for improving the specific agronomic, qualitative, and sensory parameters of other apricot varieties.

**Author Contributions:** Conceptualization, C.D.V., A.C. (Aurora Cirillo) and L.I.; methodology, A.C. (Aurora Cirillo), I.L. and C.D.V.; software, A.P. and A.C. (Aurora Cirillo); validation, C.D.V., L.I. and A.C. (Aurora Cirillo).; investigation, M.C., L.I. and I.L.; resources, A.C. (Andrea Ciervo), M.C., I.L. and L.I.; data curation, A.P.; writing—original draft preparation, A.C. (Aurora Cirillo) and L.I.; writing—review and editing, C.D.V. and A.C. (Aurora Cirillo); supervision, C.D.V. and A.P.; funding acquisition, C.D.V. All authors have read and agreed to the published version of the manuscript.

**Funding:** Trial conducted within the agreement with OP AGRICOR "Effects of treatments with biostimulants on the production, qualitative parameters and bio-active compounds of the fruits belonging to different apricot cultivars in the pedo-climatic conditions of the Piana di Sibari (CS)".

**Data Availability Statement:** Data are contained within the article.

**Acknowledgments:** The authors would like to thank OP Agricor for allowing the experimental trial to be conducted in their fields and also express gratitude to Diachem S.p.A. for their support and supply of the biostimulant products used during the trial. Special thanks are also due to Francesco Iannotta and Pietro Nicoletti for their assistance in the field.

**Conflicts of Interest:** The authors declare no conflicts of interest.

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
