# Peer review of "Optimizing Apricot Yield and Quality with Biostimulant Interventions: A Comprehensive Analysis"

_horticulturae, doi:10.3390/horticulturae10050447_

Round 1

Reviewer 1 Report

Comments and Suggestions for Authors

The manuscript investigated three products of different sources of plant biostimulants on apricot growth, yield and components, and quality. There are a lot of data, and the overall writing quality and expression quality are good. I suggest minor revisions before publication.

Abstract

(1)Line 34: does plant biostimulants belongs to fertilizers? I think some scientists disagree. This is both an academic and a market issue. Can you tell me or readers how plant biostimulants are registered in Italy, and are they regulated on the market as fertilizers or pesticide or others?

(2)Line 38: Is there a causal relationship between the dosage and their function? I don't think so. If you think so, please rewrite this sentence and explain.

(3)Add and specify previous studies on the effects of plant biostimulants on the growth, yield and quality of apricot trees.

M&M

(1)400mL/100 L should be 400 mL/100 L

(2)why called it  “Control thesis”? I know it is a control check.

(3) The water used is tap water?

(4)Tell the latitude and longitude, altitude of the site.

(5) can tell more information about the three biostimulants? Such as N, P, and K contents,and trace elements.

Results and Discussion

(1) Line 187, 3.0 should be 3

(2) Table 1: I don't think there is a statistically significant difference between the 42.75±15.62 and 31.39±8.00, please recheck.

(3) Table 1: You should list Control treatment as the first line.

(4) Figure 4, what is A, B, C, and D? Give significant difference among different treatments.

Others

The biggest regret of this paper is that it is only a one-year trial. In fact, for many international agricultural journals, “Field experiments need to be either multi-locational or multi-year and normally three at least and be accompanied by appropriate statistical analysis”. The paper only experimented one years. The expression mentioned above could be considered for inclusion in the Discussion Section or Conclusion Section of future research.

Comments on the Quality of English Language

85 in 100

Author Response

Reviewer 1

The manuscript investigated three products of different sources of plant biostimulants on apricot growth, yield and components, and quality. There are a lot of data, and the overall writing quality and expression quality are good. I suggest minor revisions before publication.

Abstract

  1. Line 34: does plant biostimulants belongs to fertilizers? I think some scientists disagree. This is both an academic and a market issue. Can you tell me or readers how plant biostimulants are registered in Italy, and are they regulated on the market as fertilizers or pesticide or others?

Dear Reviwer, in Italy according to EU Regulation 2019/1009, which came into effect on July 15, 2019, biostimulant products are fertilizers with the ability to improve tolerance to abiotic stress and qualitative characteristics, as well as nutrient use efficiency. Therefore, biostimulant products, while not being able to replace fertilizers, can be used to reduce their application enhancing its effectiveness.

  1. Line 38: Is there a causal relationship between the dosage and their function? I don't think so. If you think so, please rewrite this sentence and explain.

Dear Reviewer,

Thank you for your feedback. The term 'minute quantities' is used to distinguish these substances from fertilizers (nutrient suppliers), soil amendments (modifiers of soil physical properties), and correctives (modifiers of chemical properties), which also promote growth but are applied in larger quantities."

At the line 38 we added this sentence: infact excessive application could disrupt the plant's nutritional equilibrium and lead to adverse effects.

  1. Add and specify previous studies on the effects of plant biostimulants on the growth, yield and quality of apricot trees.

We have added at line 59-67 this sentence: “The bibliography includes several studies highlighting the use of biostimulant products of various origins to improve the productive and qualitative characteristics of apricot. Tarantino et al. [14] have demonstrated accelerated fruit ripening, enabling a greater quantity of fruit to be harvested already at the first harvest, also highlighting an improvement in the nutraceutical component. Other studies [6,15] have also shown that the application of humic acids through soil or foliar spraying positively interferes with the growth, productivity, and fruit quality of the Canino apricot cultivar observing increased fruit weight, consistency, total soluble solids, and brix-to-acidity ratio, thus promoting better fruit quality”.

  1. Rodrigues, M.; Baptistella, J.L.C.; Horz, D.C.; Bortolato, L.M.; Mazzafera, P. Organic Plant Biostimulants and Fruit Quality—A Review. Agronomy 2020, 10, 988, doi:10.3390/agronomy10070988
  2. Tarantino, A.; Lops, F.; Disciglio, G.; Tarantino, E. Effect of Plant Biostimulants on Fruit Set, Yield, and Quality Attributes of “Farbaly” Apricot Cultivar. Int. J. Agric. Biosyst. Eng. 2017, 11(8), 652-656.
  3. Eissa, F.; Fathi, M.; El-Shall, S. RESPONSE OF PEACH AND APRICOT SEEDLINGS TO HUMIC ACID TREATMENTS UNDER SALINITY CONDITION. J. Plant Prod. 2007, 32, 3605–3620, doi:10.21608/jpp.2007.208147.

M&M

  1. 400mL/100 L should be 400 mL/100 L

Done

  1. why called it  “Control thesis”? I know it is a control check.

Dear reviewer, we have changed it to a control group; furthermore, this treatment has this name to highlight the fact that all the theses with biostimulant products have been compared with a Control, namely a group of plants that have only been administered water (indeed, as a control to compare with the others that have undergone treatments with biostimulants).

  1. The water used is tap water?

The water used during the plant growth period was made available by the reclamation consortium of the Piana di Sibari and the Media Valle di Crati (CS) on a 3-day rotation.

This information has been added to line 102: “The irrigation water used during the plant growth period was made available by the reclamation consortium of the Piana di Sibari and the Media Valle di Crati (CS) on a 3-day rotation”.

  1. Tell the latitude and longitude, altitude of the site.

This information has been added at line 68: 39°43'38.5"N 16°24'18.4"E

  1. can tell more information about the three biostimulants? Such as N, P, and K contents,and trace elements.

Thank you for the suggestion, more information has been added to the lines:

76-77: The composition includes also organic carbon (C) 2.5% (25 g/l), alginic acid 10% (10 g/l), mannitol 0.4% (4 g/l).

83-84: The composition includes also total nitrogen (N) at 6%, organic nitrogen (N) at 6%, water-soluble potassium oxide (K2O) at 6%, organic carbon (C) at 18%, and amino acids at 38%.

88-91: The composition includes also total nitrogen (N) 3% (38.25 g/l), of which soluble organic nitrogen 3% (38.2 g/l), organic carbon (C) 10% (127.5 g/l), equivalent to total amino acids 18.75% (239.1 g/l), chelated micronutrients 0.5% (6.3 g/l), vitamins (B1, B2, B5, PP, C, H1) 0.1% (1.2 g/l), polysaccharides 2% (25.5 g/l), plant-origin auxinsInizio modulo

Results and Discussion

  1. Line 187, 3.0 should be 3

Done

  1. Table 1: I don't think there is a statistically significant difference between the 42.75±15.62 and 31.39±8.00, please recheck.

Dear reviewer, thank you for highlighting this point and giving us the opportunity to correct the error. In Table 1 the standard deviations of the independent groups were erroneously reported instead of the standard error of the Anova model, whose value, for each dependent measure, is the same for all groups because we used a complete experimental design and we have taken care of the correction. Moreover, unfortunately, we worked with an incomplete data matrix, so the values in the table may be different, but fortunately, the result of the analysis is the same. Thank you for raising the issue and bringing the error to our attention.

  1. Table 1: You should list Control treatment as the first line.

Dear reviewer, thank you for the suggestion. We have made the change by placing the control in the first row of all tables.

  1. Figure 4, what is A, B, C, and D? Give significant difference among different treatments.

Dear reviewer, significance has been added to this parameter. Letters B, C, D, and E correspond to different fruit diameters, specifically: B diameter ranging from 61-67 mm, C diameter ranging from 56-61 mm, D diameter ranging from 51-56 mm, and E diameter less than 56 mm.

Others

The biggest regret of this paper is that it is only a one-year trial. In fact, for many international agricultural journals, “Field experiments need to be either multi-locational or multi-year and normally three at least and be accompanied by appropriate statistical analysis”. The paper only experimented one years. The expression mentioned above could be considered for inclusion in the Discussion Section or Conclusion Section of future research.

Dear reviewer, yes, indeed, we have written the following in the final paragraph of the manuscript:Further studies are planned to confirm these findings and to determine which classes of biostimulant products are best suited for improving specific agronomic, qualitative, and organoleptic parameters in other apricot cultivars.

Reviewer 2 Report

Comments and Suggestions for Authors

Biostimulants are a very controversial topic, as their manufacturers promise astonishing results when applying extremely small doses of the product. Many times, I have come across claims that could not be confirmed. On the contrary, many studies have confirmed that the application of biostimulants is unwarranted. Therefore, I was curious about your results.

From a statistical perspective, I am missing information on meeting the requirements for using your test (Duncan). Have you verified normality and if the variances are equal? These results should be published to confirm the appropriateness of your statistical analysis.

What was the dry matter content depending on the biostimulant? Please, add this information.

Your analyses are very comprehensive and detailed, which I appreciate. The results seem to be fine if the method is confirmed to be correct. I find the 11 kg difference in production (Table 1) hard to believe, but that is my problem.

Increased yields are likely to be associated with higher demands on crop nutrition. Will you also analyse nutrient uptake and calculate fertilizer adjustment?

The article needs typographical editing. Please go through the whole article and check the typography in detail.

Results from one year are not standard in the field of nutrition. Even for the use of biostimulants, it would be useful to publish results from multiple years. Do you plan to do so?

Author Response

Reviewer 2

Biostimulants are a very controversial topic, as their manufacturers promise astonishing results when applying extremely small doses of the product. Many times, I have come across claims that could not be confirmed. On the contrary, many studies have confirmed that the application of biostimulants is unwarranted. Therefore, I was curious about your results.

From a statistical perspective, I am missing information on meeting the requirements for using your test (Duncan). Have you verified normality and if the variances are equal? These results should be published to confirm the appropriateness of your statistical analysis.

Dear reviewer, thank you for your consideration, after verifying both that all the measures were normally distributed through the Kolmogorov-Smirnov test, and the homogeneity of the variances through the Levene's test, a one-way ANOVA was applied for each measure. Below are the analyses conducted:

Tests of Normality

Kolmogorov-Smirnova

Statistic

df

Sig.

Production per Plant (kg)

0.157

24

0.128

Yield Efficiency (kg/cm2)

0.154

24

0.146

First  Harvest

0.121

32

,200*

What was the dry matter content depending on the biostimulant? Please, add this information.

Dear Reviewer,

Unfortunately, we do not have this data. Nevertheless, we sincerely appreciate your valuable input, and we will certainly take into account the analysis of this parameter in the subsequent trial.

Your analyses are very comprehensive and detailed, which I appreciate. The results seem to be fine if the method is confirmed to be correct. I find the 11 kg difference in production (Table 1) hard to believe, but that is my problem.

Dear reviewer, thank you for highlighting this point and giving us the opportunity to correct the error. In Table 1 the standard deviations of the independent groups were erroneously reported instead of the standard error of the Anova model, whose value, for each dependent measure, is the same for all groups because we used a complete experimental design and we have taken care of the correction. Moreover, unfortunately, we worked with an incomplete data matrix, so the values in the table may be different, but fortunately, the result of the analysis is the same. Thank you for raising the issue and bringing the error to our attention.

Increased yields are likely to be associated with higher demands on crop nutrition. Will you also analyse nutrient uptake and calculate fertilizer adjustment?

Dear Reviewer, we would like to emphasize that the fertilization plan is conducted based on the nutrient removals from the previous year. Thank you for the suggestion.

The article needs typographical editing. Please go through the whole article and check the typography in detail.

Dear reviewer, the authors have checked and improved the typography of the article.

Inizio modulo

Results from one year are not standard in the field of nutrition. Even for the use of biostimulants, it would be useful to publish results from multiple years. Do you plan to do so?

Dear reviewer, yes, indeed, we have written the following in the final paragraph of the manuscript:Further studies are planned to confirm these findings and to determine which classes of biostimulant products are best suited for improving specific agronomic, qualitative, and organoleptic parameters in other apricot cultivars
